# TABULAR DEEP-SMOTE:
# A SUPERVISED AUTOENCODER-BASED MINORITY-OVERSAMPLING TECHNIQUE FOR CLASS-IMBALANCED TABULAR CLASSIFICATION

## ABSTRACT

Class imbalance, present in many real-world tabular datasets, may cause machine-learning models to under-classify minority samples, which are often highly significant. This work proposes a new oversampling method called Tabular Deep-SMOTE (TD-SMOTE), which harnesses the class labels to improve synthetic sample generation via autoencoders. The method is based on oversampling in an alternative space shaped by a metric-learning loss. Such spaces tend to be more semantic and obtain higher class separation and density, which improves the quality of samples generated by linear interpolations over the observed minority samples. In addition, we propose a synthetic samples filtering scheme based on the decision boundary of a pre-trained tabular classifier to guarantee the quality of synthetic samples. Compared to common and leading oversampling methods, the method achieves improved classification performance in an extensive set of experiments that includes over 36 publicly available datasets.

## 1 INTRODUCTION

Class imbalances are common in real-world tabular datasets and occur across many domains (Yu et al., 2012; Tek et al., 2010; Horta et al., 2008; Chan & Stolfo, 1998). Class imbalances tend to lead to predictive biases in standard classifiers, resulting in poor performance over classes with fewer instances (He & Garcia, 2009). This tendency is detrimental as less frequent classes are often vital events, such as system failure or a rare disease. As a result, the class imbalance problem has been considered important for many years and is intensively researched (He & Garcia, 2009).

Several methods have been proposed to improve classifiers in the class-imbalance setting (Johnson & Khoshgoftaar, 2019). These methods can be categorized into two groups: algorithmic-oriented and data-oriented approaches. The algorithmic-oriented approaches modify the classifier to handle the class imbalance better (Lee et al., 2021; Jo & Japkowicz, 2004). The most basic example of such an approach would be loss re-weighting, namely increasing the minority class weight in the loss function used during the classifier training. The data-oriented approaches modify the training set such that the resulting set has a lower class-imbalance ratio. The data-oriented approach can be categorized into two sub-categories - under-sampling of majority samples (Tomek, 1976; Barua et al., 2013a) and over-sampling of minority samples (More, 2016; Batista et al., 2004). The most basic example of such an approach would be random oversampling, in which random minority samples in the training set are duplicated.

*Synthetic Minority Oversampling Technique* (SMOTE)(Chawla et al., 2002) was one of the first methods to propose dataset balancing by adding synthetic minority samples. The synthetic samples are generated by interpolating minority samples and their nearest minority neighbors. Throughout the years, more than 100 variations of the original SMOTE algorithm were proposed (More, 2016). Several of these variations are based on a SMOTE oversampling in an alternative space (Wang et al., 2006; Tang & Chen, 2008; Gu et al., 2009; Pérez-Ortiz et al., 2016). In these methods; the data is mapped to a desired alternative space, oversampled, and inversely mapped back to the original space. In recent years, neural-network-based models achieved substantial success in various fields

(Alam et al., 2020). This success led to several autoencoder (AE) based oversampling techniques that follow the same oversampling principle in the latent space of a trained AE (Bellinger et al., 2015; Darabi & Elor, 2021). *Denoising Autoencoder-Based Generative Oversampling* (DEAGO) (Bellinger et al., 2015) demonstrates promising results by training a Denoising-AE over the minority samples. *Tabular AutoEncoder Interpolator* (TAEI) (Darabi & Elor, 2021) went a step further by training AE-based models over the entire training set (minority and majority). In addition, TAEI adds support for categorical features as interpolating in the latent space avoids interpolating over categories (which is a non-trivial task). One key limitation of DEAGO and TAEI is that they do not use the available labels to shape the AE's latent space.

This work presents a new oversampling scheme called *Tabular Deep-SMOTE* or TD-SMOTE (outline in Fig. 1). Similarly to DEAGO and TAEI, the proposed method is based on interpolations in a latent space of an ad-hoc trained AE (①in Fig. 1). Contrary to them, our method trains the AE in a supervised fashion; namely, it utilizes the labels to train the AE. A metric learning loss is applied over the latent representations of minority and majority samples in the training stage. Such a loss encourages the learned latent space to be highly class-separated, class-dense, and with Euclidean distances that better reflect the similarities (semantic latent space). Thus, interpolations of minority sample embeddings generated by SMOTE are closer to high density minority regions.

In addition, our method introduces into the AE framework an importance-oversampling scheme that prioritizes oversampling near class domain boundaries (②in Fig. 1). This approach has been proven to improve down-stream classifiers (More, 2016) and is used by several oversampling techniques (Han et al., 2005; Gradstein et al., 2022; Barua et al., 2013b). TD-SMOTE adapts a priority weight algorithm used in *Proximity Weighted Synthetic Oversampling* (ProWSyn)(Barua et al., 2013b). The adaptation associates with each minority sample a weight or probability reflecting its importance (hence, the term importance-oversampling). At the oversampling stage, a minority latent representation is sampled according to the set of probabilities, and then it is interpolated with one of its $k$-nearest-neighbors (NNs), similar to SMOTE. We name this SMOTE variation Weighted-SMOTE and the overall oversampling approach importance-oversampling.

The third and last element of our method is a synthetic sample filtering scheme (③in Fig. 1). The metric-learning loss added over the AE's latent space encourages class separation. Nevertheless, a fully class-separated space is not guaranteed, and interpolations between minority representations may cross minority-majority domain boundaries. By filtering synthetic samples that are remote from high density minority regions, the overall quality of the oversampled training set is improved (Sáez et al., 2015; Lee et al., 2015). We introduce a new approach for filtering that uses a baseline classifier trained over the original data. The motivation for using a classifier stems from experimental results, provided in Section 4, which demonstrate that current state-of-the-art classifiers, such as Catboost (Prokhorenkova et al., 2017), achieve reasonable to good results even over highly class-imbalanced datasets. The filtering scheme is based on creating a minimal threshold score below which synthetic samples are discarded. The threshold is determined based on the interpolated minority samples and the interpolation factor (i.e., distance from each point). If the score of the synthetic sample degrades below this local threshold, the sample is considered to have crossed class boundaries and is discarded.

We provide the following contributions:

1. Demonstrate that by training with a metric-learning loss, class information (labels) can be utilized to learn a latent space that is more suitable than the original space for oversampling with SMOTE (Section 3.1.1);

2. Introducing into the AE oversampling framework an importance-oversampling scheme (Section 3.2.1);

3. Present a unique approach to filter synthetic samples based on a trained baseline-classifier and local thresholding (Section 3.2.2);

4. Demonstrate that the TD-SMOTE model, which comprises all three parts - training with a metric-learning loss, importance-oversampling, and baseline classifier filtering - achieves better prediction quality in downstream classification tasks compared to common and leading schemes, such as SMOTE (Chawla et al., 2002), SMOTE-IPF (Sáez et al., 2015) and ProWSyn (Barua et al., 2013b).

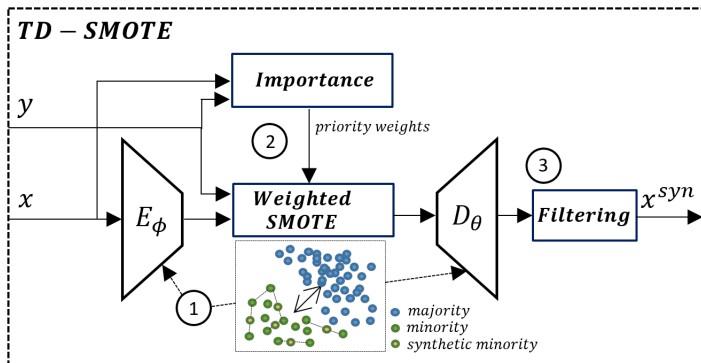

Figure 1: TD-SMOTE outline - the model gets as inputs the data samples, $x$, and their associated labels, $y$. New minority samples, $x^{syn}$, are generated and filtered using the three components of the model - the ad-hoc trained autoencoder, importance oversampling and filtering based on the decision boundary of a pre-trained tabular classifier.

## 2 RELATED WORK

*Synthetic Minority Oversampling Technique* (SMOTE)(Chawla et al., 2002) was one of the first methods to propose dataset balancing by adding synthetic minority samples. The synthetic samples are generated by interpolating minority samples and their nearest minority neighbors. Interpolating based solely on the nearest minority neighbors is a local operation that does not take into account the entire minority or majority distribution. Ignoring these distributions may render the oversampling either less effective when oversampling is done in an already overly populated minority domain or even detrimental when oversampling crosses into the majority domain. This drawback, among others, led to the proposal of more than 100 variations of the original SMOTE algorithm (More, 2016). TD-SMOTE, which can be categorized as an advanced SMOTE variation, is based on three principles - (a) importance-oversampling, (b) filtering of the generated synthetic samples, and (c) oversampling in an alternative space.

**Importance-oversampling**, as referred to in this work, is the principle of prioritizing oversampling based on certain minority samples over others. This approach is used in oversampling techniques prioritizing sampling near minority-majority domain boundaries (Han et al., 2005; He et al., 2008; Li et al., 2014; Gradstein et al., 2022). Seeing that this principle demonstrates improvements in down-stream classifiers (More, 2016), TD-SMOTE adapts into the AE framework an importance estimation scheme presented in *Proximity Weighted Synthetic Oversampling* (ProWSyn)(Barua et al., 2013b). ProWSyn partitions the minority samples into clusters according to their proximity to majority class samples. A weight is assigned to each cluster of minority samples, such that clusters closer to the majority class have higher weights and vice versa. These weights determine how many data points will be generated from each cluster. Synthetic samples are then generated within each cluster via SMOTE.

**Synthetic Sample Filtering** is another approach that has proven beneficial in minority oversampling (More, 2016). Such oversampling techniques use filtering schemes to filter out synthetic samples that are estimated to be remote from the minority distribution. Two high-performing oversampling schemes that use such filtering are - *SMOTE Iterative-Partitioning Filter* (SMOTE-IPF) (Sáez et al., 2015) and *Oversampling Technique with Rejection* (OTR) (Lee et al., 2015). SMOTE-IPF oversamples the dataset with SMOTE and uses a noise-filtering scheme named *Iterative-Partitioning Filter* (IPF) (Khoshgoftaar & Rebours, 2007) combined with a trained decision tree classifier to filter out synthetic samples that appear non-consistent with the original data. OTR oversamples the dataset with SMOTE and uses a $k$-NNs classifier to filter out synthetic samples for which most of their nearest neighbors are non-minority samples. Our proposed method follows these techniques by proposing a new filtering scheme based on a pre-trained tabular classifier.

**Oversampling in an Alternative Space** is the third principle on which TD-SMOTE is based. Several oversampling techniques oversample in a space alternative to the original features. In these methods, the data is mapped to the alternative representation, oversampling is done, and the gener-

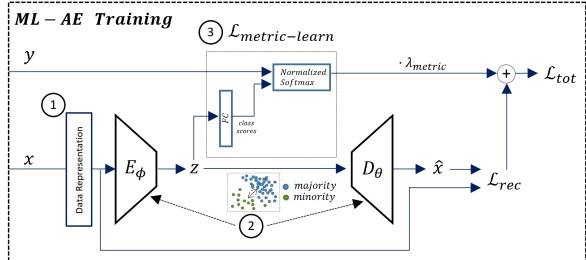

Figure 2: ML-AE training outline - includes (1) the data representation stage, (2) the encoder-decoder pair, and (3) the metric-learning loss in the latent representation. The metric learning loss consists of a fully connected layer and the Normalized SoftMax loss (Zhai & Wu, 2018). Applying the metric learning loss results in better class separation and denser class cluster representation where Euclidean distances reflect similarity, thus improving the effectiveness of SMOTE.

ated synthetic samples are then inverse-mapped back to the original data space. Such methods include dimensionality reduction via PCA (Tang & Chen, 2008), Locally Linear Embedding (Roweis & Saul, 2000; Wang et al., 2006) and Isomap (Tenenbaum et al., 2000; Gu et al., 2009). In recent years, neural networks achieved substantial success in various fields (Alam et al., 2020). This success led to several autoencoder (AE) based oversampling techniques which follow the same principle by using the AE's latent space as the alternative space for oversampling (Bellinger et al., 2015; 2016; Darabi & Elor, 2021). As detailed in Section 1, these methods do not use the labels for training the AE. One exception is *DeepSMOTE* (Dablain et al., 2021) - an AE-based oversampling method for image datasets that uses labels with an intra-class permutation loss *"designed to insert variance into the encoding/decoding process and therefore obviate the need for a discriminator"*. The proposed permutation loss proved ineffective in this work's scope (see Appendix A); therefore, we have introduced an alternative metric-learning loss.

## 3 TD-SMOTE

This section presents the TD-SMOTE method. The first stage involves feature preprocessing. This stage includes data imputation, feature scaling, and representation of categorical features. Further details are provided in Appendix B. The second stage involves training the metric-learning AE that we denote by ML-AE. The ML-AE training procedure, architecture, and the chosen metric learning loss are presented in Section 3.1. The third and last stage involves oversampling the dataset using the trained ML-AE (Section 3.2).

### 3.1 Model Training

The model training stage uses the preprocessed features (see (1) in Fig. 2) extracted as described in Appendix B. Each sample in the batch, denoted by $x$, is encoded into its representation in the latent space, denoted by $z$, via the encoder $E_\phi : \mathcal{X} \to \mathcal{Z}$. The latent representation is then decoded into the reconstructed sample, denoted by $\hat{x}$, via the decoder $D_\psi : \mathcal{Z} \to \mathcal{X}$. The total loss per sample, denoted by $\mathcal{L}_{tot}$ is a combination of the reconstruction loss, denoted by $\mathcal{L}_{rec}$, and the metric-learning loss over $z$, denoted by $\mathcal{L}_{metric\_learn}$. Balancing $\mathcal{L}_{rec}$ with $\mathcal{L}_{metric\_learn}$ is done using a factor denoted by $\lambda_{metric}$. The training algorithm is described in Algorithm 1. The ML-AE architecture (see (2) in Fig. 2), which comprises both the encoder and decoder, is described in Appendix C. The metric-learning loss (see (3) in Fig. 2) is described in Section 3.1.1.

### 3.1.1 Metric Learning Loss Choice

*Metric Learning* aims at learning a metric from high-dimensional data such that the distance between two data samples, as measured by the metric, reflects their semantic similarity (Lu et al., 2017; Kaya & Bilge, 2019). In other words, it aims at learning a class discriminative representation such that embeddings of data samples from the same class are encouraged to be similar. In contrast, embeddings of data samples from different classes should be dissimilar. We hypothesize that oversampling in a latent space shaped by a metric learning loss can improve the quality of synthetically generated samples for several reasons:

---

**Algorithm 1** ML-AE Training

---

**Input**: $(x_{train}, y_{train})$ - data samples and their labels.
**Output**: $\theta$ - model parameters (encoder and decoder parameters $\phi$ and $\psi$, respectively, and the metric-learning loss's fully connected layer weights)

1: Initialize parameters: $\theta \leftarrow$ random parameters
2: Data representation:
3: $x'_{train} = \begin{cases} onehot(x_{i,train}) & categorical\ feature \\ x_{i,train} & numeric\ feature \end{cases}$
4: **repeat**
5:     **for each** batch **do**:
6:         $x_{batch}, y_{batch} \leftarrow$ next batch of $(x'_{train}, y_{train})$
7:         $z_{batch} = E_{\phi}(x_{batch})$
8:         $\hat{x}_{batch} = D_{\psi}(z_{batch})$
9:         $\mathcal{L}_{rec} =$
10: $mean \begin{cases} cross\_entropy(\hat{x}_{i,batch}, x_{i,batch}) & categorical\ feature \\ l_2(\hat{x}_{i,batch}, x_{i,batch}) & numeric\ feature \end{cases}$
11:         $\mathcal{L}_{tot} =$
12: $\mathcal{L}_{rec} + \lambda_{metric}\mathcal{L}_{metric-learn}(z_{batch}, y_{batch})$             ▷ See eq.1
13:         $\theta \leftarrow AdamOptimizer(\theta, \nabla_{\theta}\mathcal{L}_{tot})$
14:     **end for**
15: **until** $\mathcal{L}_{tot}$ convergence (or maximal number of epochs)
16: **return** $\theta$

---

1. *Improving the similarity metric between minority samples*: SMOTE generates synthetic samples by interpolating minority samples with their $k$-NNs. Interpolation with the $k$-NNs is based on the assumption that Euclidean distances reflect similarity. Metric learning may provide a framework in which a more semantic similarity metric is learned and used as an alternative for Euclidean distances in the original data space. Using an alternative similarity measure has proven beneficial for interpolation-based oversampling (Bej et al., 2019; 2021b).

2. *Class separability (contrastive)*: interpolation-based oversampling methods may generate samples that cross into the majority class domain (see illustration in Fig.4 Appendix D). Metric learning losses encourage class separation in the learned latent space. This may alleviate the risk of generating false minority samples.

3. *Class density (tightness)*: metric learning lossses encourage intra-class representations to be close. As a result, the latent space is denser, and interpolations may represent much higher probability minority samples once they are decoded back to the original space (see illustration in Fig.5 in Appendix D). This is beneficial as the downstream learned classifier may be more accurate when trained using synthetic samples that are closer to high density minority regions.

We have explored several metric-learning losses for our purpose[1], namely, improving the oversampling effectiveness for downstream classification tasks. In the scope of this evaluation, a linear layer followed by the Normalized Softmax loss (Zhai & Wu, 2018) demonstrated the best performance. For the rest of our experiments, we use this loss. Another justification for using the Normalized Softmax loss Boudiaf et al. (2020) is that it avoids the increased complexity of mining pairs or triplets, which are commonly used by other metric losses (Chopra et al., 2005; Schroff et al., 2015).

The high-class imbalance may be detrimental when using a standard metric learning loss that is not accustomed to handling the class imbalance (Gautheron et al., 2019; Gui & Zhang, 2021; Wang et al., 2021). To alleviate the effect of class imbalance, we apply loss re-weighting by class. Each class is re-weighted with a factor, $w_0^{RW}$ for the majority class and $w_1^{RW}$ for the minority one. We

---

[1]based on the PyTorch Metric Learning library- https://kevinmusgrave.github.io/pytorch-metric-learning/

chose the following common factors:

$$imb\_ratio \stackrel{\text{def}}{=} \frac{\#majority\ samples}{\#minority\ samples} \;\; ; \;\; w_0^{RW} := \sqrt{\frac{1}{imb\_ratio}} \;\; ; \;\; w_1^{RW} := \sqrt{imb\_ratio}$$

It is worth mentioning that in the scope of our experiments re-weighting the reconstruction loss, $\mathcal{L}_{rec}$, by class did not demonstrate any significant improvement and was not used. The class re-weighted Normalized Softmax is the loss used for the metric learning purpose (see Algorithm 1):

$$\mathcal{L}_{\textbf{metric−learn}}(\mathbf{z}, \mathbf{y}) = -\sum_{i=1}^{m} w_{y_i}^{RW} log \frac{e^{cos(w'_{y_i}, z_i)}}{e^{cos(w'_0, z_i)} + e^{cos(w'_1, z_i)}} \tag{1}$$

Where $\mathbf{z}$ are the latent representations of the batch, $\mathbf{y}$ are the batch labels, $m$ is the batch size, $cos$ denotes the cosine similarity measure, and $w'_0$ and $w'_1$ are the linear layer weights associated with the majority and minority classes, respectively.

## 3.2 OVERSAMPLING

The minority oversampling is done by running the following steps: (a) the minority samples are encoded using the trained encoder, (b) the encodings are oversampled in the latent space using importance-oversampling (Section 3.2.1), (c) the interpolations are decoded using the trained decoder and, finally, (d) synthetic samples that are remote from high density minority regions are filtered out (Section 3.2.2). The complete oversampling procedure is described in Algorithm 2.

---

**Algorithm 2** Oversample

**Inputs**: trained ML-AE, $(x, y)$ - imbalanced dataset, $\lambda^{os}$ - oversample ratio, $classifier\_type$ - classifier type used for the baseline filtering (e.g. SVM / CatBoost)
**Output**: $(x^{os}, y^{os})$ - oversampled dataset
1: $num\_required \leftarrow sum(y == 0) \cdot \lambda^{os} - sum(y == 1)$
2: $weights \leftarrow$ GetImportance$(x, y)$           ▷ Algorithm 3
3: $baseline\_classifier \leftarrow$ train a baseline classifier of type $classifier\_type$ over $(x, y)$
4: $x_{syn}^{final} \leftarrow [\,]$
5: $x' \leftarrow \begin{cases} onehot(x_i) & i^{th}\ feature\ is\ categorical \\ x_i & i^{th}\ feature\ is\ numeric \end{cases}$
6: $z \leftarrow E_\phi(x')$
7: $z_{syn} \leftarrow$ WeightedSMOTE$(z, weights, num\_required)$     ▷ Algorithm 4
8: $x'_{syn} \leftarrow D_\psi(z_{syn})$
9: $x_{syn} \leftarrow \begin{cases} argmax(x'_i) & i^{th}\ feature\ is\ categorical \\ x'_i & i^{th}\ feature\ is\ numeric \end{cases}$
10: $x_{syn} \leftarrow$ Filter$(baseline\_classifier, x_{syn}, ...)$       ▷ Algorithm 5
11: $x_{syn}^{final} \leftarrow x_{syn}^{final} \,||\, x_{syn}$         ▷ concatenate new samples
12: **if** $|x_{syn}| < num\_required$ **then**:
13:    $num\_required \mathrel{-}= |x_{syn}|$
14:    **go to** 7
15: **end if**
16: $x^{os} = x \,||\, x_{syn}^{final}$
17: $y^{os} = y \,||\, 1^{num\_required}$           ▷ minority label is 1
18: **return** $(x^{os}, y^{os})$

---

### 3.2.1 IMPORTANCE-OVERSAMPLING

Oversampling near minority-majority domain boundaries has proven effective for downstream classification tasks (More, 2016). Many methods, such as borderline-smote (Han et al., 2005), ADASYN (He et al., 2008), and ProWSyn (Barua et al., 2013b), are partly or entirely based on it. We adopt an importance sampling method based on the one used in ProWSyn (Barua et al., 2013b). The algorithm comprises two parts: *GetImportance* presented in Algorithm 3 and *WeightedSMOTE* presented in Algorithm 4. Both parts are used separately in the oversampling stage, as shown in Algorithm 2.

*GetImportance* assigns each minority sample an importance value that represents the sample's probability of being used in generating a new synthetic sample. The importance value assigned to each minority sample is determined according to the minority sample's proximity to the majority samples. The higher the proximity, the higher the importance. The method is presented in detail in Algorithm 3, accompanied by an illustration in Fig. 6 in Appendix E.

*WeightedSMOTE*, the second part of the importance oversampling scheme, is a SMOTE extension that supports prioritizing specific minority samples according to the importance distribution provided by *GetImportance*. WeightedSMOTE selects a minority sample according to the importance values and then interpolates it with one of its $k$-NNs to generate a new synthetic sample. This scheme is similar to SMOTE except for the fact that SMOTE chooses a minority sample uniformly, while WeightedSMOTE samples according to the importance distribution. The method is detailed in Algorithm 4 in Appendix F.

---

**Algorithm 3** GetImportance

---

**Input**: $(x, y)$ - labeled dataset
**Output**: $weights$ - importance weights for minority samples

1: $k \leftarrow 5$ ▷ used as the default value
2: $M \leftarrow x[y == 0]$ ▷ set of majority samples
3: $m \leftarrow x[y == 1]$ ▷ set of minority samples
4: $weights \leftarrow zeros(size(m))$ ▷ initialize with zeros
5: **for** $level$ in $[1, ..., 4]$ **do**
6:     $k\_neighbors \leftarrow set()$ ▷ does not include duplicates
7:     **for** $maj\_sample$ in $M$ **do**
8:         $k\_neighbors$.add($k$ nearest-neighbors of $maj\_sample$ from $m$)
9:     **end for**
10:     $weights[k\_neighbors] \leftarrow exp^{level-1}$
11:     $m \leftarrow m \setminus k\_neighbors$ ▷ update minority set
12: **end for**
13: $weights[m] \leftarrow exp^{5-1}$ ▷ assign the remaining minority samples the weight of the last level (5)
14: $weights \leftarrow weights/sum(weights)$ ▷ transform weights into probabilities
15: **return** $weights$

---

### 3.2.2 CLASSIFIER FILTERING

Although the latent space is trained to be contrastive, the inherent problem of synthetic samples crossing into the majority domain may still exist. This results from the fact that perfect class separation is rarely achievable. We proposed another effective method to address this issue - baseline classifier filtering. This approach filters the synthetic samples using a baseline classifier trained over the original dataset. The underlying assumption is that the classifier has a somewhat reasonable ability to classify the data regardless of the imbalance and can be used as a "sanity" check for the quality of new synthetic samples.

Most classifiers, including Catboost (Prokhorenkova et al., 2017) and SVM used in our experiments, can estimate the probability of a given sample belonging to each one of the classes. We refer to the probability of a given sample belonging to the minority class as the sample's score. Scores may range from 0 to 1, representing complete confidence in the sample being a majority or minority instance. We use the trained baseline classifier to assign each synthetic sample a score in the oversampling stage. Synthetic samples with a score less than a predetermined threshold are discarded, and the oversampling continues until the required amount of synthetic samples is generated. The threshold presented in eq. 2, is determined according to the two scores of the minority samples being interpolated, denoted by $base\_score$ and $nhbr\_score$, and the interpolation factor, denoted by $\lambda$. Because these two scores may have different scales (e.g., 0.05 and 0.5), the threshold is based on a logarithmic interpolation, namely, $\log x = \log x_1 + \lambda(\log x_2 - \log x_1) \rightarrow x = x_1^{1-\lambda} x_2^{\lambda}$.

$$Threshold\ Score\ (\lambda, base\_score, nhbr\_score) = \frac{(base\_score)^{1-\lambda}(nhbr\_score)^{\lambda}}{margin} \quad (2)$$

The $margin$ in eq. 2 is added as a leeway margin to avoid over-filtering. The chosen default value is 2. The proposed threshold is relatively simple yet effective (as demonstrated in the ablation studies in Appendix M). The filtering algorithm is detailed and visually demonstrated in Appendix G.

## 4 EXPERIMENTAL RESULTS

A comparison is performed over 36 publicly available tabular datasets[2] [3]. Experiments include the state-of-the-art CatBoost (Prokhorenkova et al., 2017) and SVM as down-stream classifiers and several evaluation metrics - Average Precision (AP), F1, and ROC-AUC. Further details are provided in Appendix I.

We evaluate the performance of several leading oversampling techniques on the *purely numeric datasets*, namely, we use (More, 2016) - borderline smote (bsmote) (Han et al., 2005), Polynomial-Fit SMOTE (poly_fit) (Gazzah & ESSOUKRI BEN AMARA, 2008), Prowsyn (Barua et al., 2013b) and SMOTE IPF (Sáez et al., 2015). CTGAN (Xu et al., 2019), a state-of-the-art tabular generative model, is also included in the evaluation. The classification results with a down-stream CatBoost classifier (Prokhorenkova et al., 2017) are presented in Table 1. TD-SMOTE achieves the lowest AP rank among the used baselines (Fig. 3).

For *datasets with categorical features* we only benchmark a smaller set of methods that support the synthetic generation of categorical values. These include - SMOTE Nominal-Continuous and SMOTE Nominal (SMOTENC / SMOTEN) (Chawla et al., 2002) and CTGAN. The CatBoost classifier supports categorical features without additional modification or alternative data representation. Classification results are presented in Table 2. The results and overall rank, as presented in Fig. 3, indicate that TD-SMOTE can successfully support categorical features.

Table 1: Classification AP results for numerical datasets using CatBoost

| | no oversample | smote | bsmote1 | bsmote2 | poly_fit | prowsyn | smote_ipf | random oversample | loss reweight | ctgan | td-smote |
|---|---|---|---|---|---|---|---|---|---|---|---|
| glass-0-1-6_vs_2 | 0.434 | 0.439 | 0.384 | 0.455 | 0.177 | **0.479** | 0.375 | 0.380 | 0.376 | 0.315 | 0.379 |
| glass2 | 0.234 | 0.276 | 0.256 | 0.331 | 0.250 | 0.245 | 0.261 | 0.237 | **0.350** | 0.252 | 0.279 |
| glass4 | 0.809 | 0.700 | 0.467 | **1.000** | 0.411 | 0.533 | 0.589 | 0.639 | 0.806 | 0.444 | 0.756 |
| page-blocks-1-3_vs_4 | 0.967 | 0.967 | 0.967 | **1.000** | 0.943 | 0.943 | 0.967 | **1.000** | **1.000** | **1.000** | 0.967 |
| yeast-0-5-6-7-9_vs_4 | 0.783 | 0.737 | 0.682 | 0.707 | **0.796** | 0.667 | 0.769 | 0.782 | 0.674 | 0.692 | 0.634 |
| yeast-1_vs_7 | **0.586** | 0.224 | 0.350 | 0.361 | 0.159 | 0.253 | 0.249 | 0.311 | 0.288 | 0.518 | 0.523 |
| yeast-1-2-8-9_vs_7 | 0.209 | 0.201 | 0.214 | 0.209 | 0.093 | 0.125 | 0.205 | 0.207 | 0.216 | 0.148 | **0.254** |
| yeast-1-4-5-8_vs_7 | 0.085 | 0.371 | **0.454** | 0.300 | 0.104 | 0.384 | 0.393 | 0.281 | 0.274 | 0.072 | 0.118 |
| yeast-2_vs_4 | 0.925 | 0.948 | 0.900 | 0.861 | 0.903 | 0.943 | 0.915 | 0.911 | 0.946 | **0.975** | 0.927 |
| yeast-2_vs_8 | 0.524 | 0.528 | 0.317 | 0.232 | 0.526 | 0.526 | 0.522 | 0.533 | 0.521 | **0.543** | 0.535 |
| yeast4 | 0.417 | 0.385 | 0.324 | 0.264 | 0.443 | 0.386 | 0.384 | 0.431 | 0.393 | 0.429 | **0.502** |
| yeast5 | 0.905 | 0.800 | 0.877 | 0.928 | 0.900 | 0.808 | 0.855 | 0.849 | 0.815 | **0.960** | 0.932 |
| yeast6 | 0.722 | 0.734 | 0.649 | 0.681 | 0.713 | 0.731 | 0.669 | 0.701 | 0.754 | 0.690 | **0.772** |
| ecoli | **0.564** | 0.415 | 0.320 | 0.321 | 0.471 | 0.392 | 0.418 | 0.488 | 0.438 | 0.508 | 0.463 |
| letter_img | 0.981 | 0.986 | 0.979 | 0.972 | 0.985 | 0.986 | 0.989 | **0.994** | 0.993 | 0.979 | 0.987 |
| libras_move | **0.798** | 0.645 | 0.677 | 0.630 | 0.657 | 0.661 | 0.652 | 0.629 | 0.650 | 0.742 | 0.637 |
| mammography | 0.699 | 0.707 | 0.672 | 0.614 | 0.717 | 0.679 | 0.708 | **0.723** | 0.702 | 0.581 | 0.690 |
| ozone_level | **0.351** | 0.323 | 0.304 | 0.282 | 0.275 | 0.349 | 0.301 | 0.294 | 0.302 | 0.217 | 0.239 |
| pen_digits | **1.000** | **1.000** | 1.000 | 0.999 | 1.000 | 1.000 | **1.000** | **1.000** | **1.000** | 1.000 | **1.000** |
| satimage | 0.755 | 0.772 | 0.735 | 0.729 | 0.781 | 0.780 | **0.796** | 0.777 | 0.739 | 0.702 | 0.783 |
| spectrometer | 0.989 | 0.963 | 0.972 | **1.000** | 0.989 | **1.000** | 0.930 | 0.977 | 0.980 | 0.989 | **1.000** |
| us_crime | 0.388 | **0.476** | 0.368 | 0.381 | 0.432 | 0.442 | 0.417 | 0.420 | 0.420 | 0.389 | 0.474 |
| webpage | 0.698 | 0.590 | 0.665 | 0.678 | 0.692 | 0.672 | 0.607 | 0.663 | 0.663 | **0.707** | 0.684 |
| wine_quality | **0.323** | 0.209 | 0.182 | 0.117 | 0.119 | 0.146 | 0.200 | 0.279 | 0.228 | 0.169 | 0.162 |
| yeast_me2 | 0.601 | 0.654 | 0.584 | 0.525 | **0.704** | 0.657 | 0.679 | 0.666 | 0.695 | 0.606 | 0.621 |
| yeast_ml8 | **0.138** | 0.088 | 0.100 | 0.108 | 0.109 | 0.099 | 0.099 | 0.111 | 0.083 | 0.106 | 0.108 |
| coil_2000 | 0.169 | **0.190** | 0.164 | 0.163 | 0.181 | 0.184 | 0.167 | 0.154 | 0.167 | 0.173 | 0.180 |
| oil | 0.347 | 0.463 | 0.425 | 0.459 | 0.563 | 0.410 | 0.342 | 0.722 | 0.653 | 0.477 | **0.764** |
| optical_digits | **0.996** | 0.992 | 0.993 | 0.988 | 0.994 | 0.994 | 0.995 | 0.990 | 0.993 | 0.994 | 0.993 |
| arrhythmia | **0.877** | 0.810 | **0.877** | 0.629 | 0.810 | **0.877** | 0.810 | 0.810 | **0.877** | 0.810 | **0.877** |
| car_eval_34 | 0.746 | 0.991 | 0.968 | 0.979 | 0.906 | **1.000** | 0.996 | 0.990 | 0.885 | 0.775 | 0.894 |
| #Best | 9 | 3 | 2 | 3 | 2 | 4 | 2 | 4 | 4 | 5 | 7 |
| #Worst | 2 | 2 | 3 | 9 | 5 | 1 | 2 | 2 | 1 | 4 | 1 |

---

[2]https://sci2s.ugr.es/keel/imbalanced.php?order=ins#sub30

[3]https://imbalanced-learn.org/stable/datasets/index.html

Table 2: Classification AP results for categorical and mixed datasets using Catboost

|  | no oversample | smoten / smotenc | random oversample | loss reweight | ctgan | td-smote |
|---|---|---|---|---|---|---|
| abalone | 0.400 | 0.405 | 0.418 | **0.488** | 0.380 | 0.434 |
| abalone_19 | 0.050 | 0.016 | 0.045 | 0.025 | **0.053** | 0.027 |
| sick_euthyroid | 0.878 | 0.879 | 0.882 | 0.879 | 0.812 | **0.898** |
| thyroid_sick | 0.961 | 0.958 | 0.964 | 0.964 | 0.959 | **0.966** |
| solar_flare_m0 | 0.101 | 0.122 | 0.123 | 0.104 | **0.190** | 0.187 |
| #Best | 0 | 0 | 0 | 1 | 2 | 2 |
| #Worst | 1 | 2 | 0 | 0 | 2 | 0 |

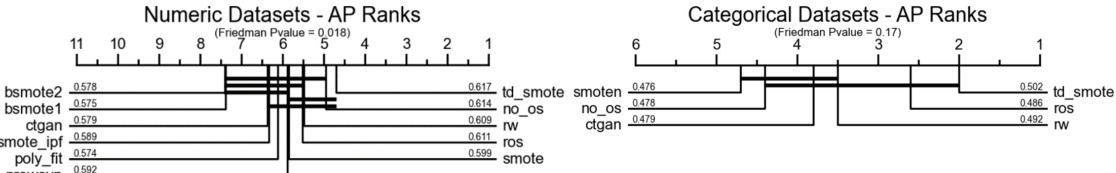

Figure 3: Critical Difference Diagram - Wilcoxson significance analysis over datasets with numeric features (left figure, based on Table 1) and datasets with categorical features (right figure, based on Table 2). Methods connected via a bold bar indicate that performances are not significantly different, namely that $p > 0.05$. Average AP scores are presented above lines.

### 4.1 ADDITIONAL EXPERIMENTS

A toy example using a synthetic dataset is provided in Appendix H. The oversampled datasets demonstrate the difference between oversampling techniques and TD-SMOTE's advantage over them.

A performance evaluation may be done using several different measures. To further demonstrate the effectiveness of TD-SMOTE, the same evaluations presented in this section were done using the F1-scores and ROC-AUC values. F1-scores are presented in Appendix K.1 and ROC-AUC values in Appendix K.2. Classification results with a RBF-Kernel SVM are presented in Appendix L. In all cases, TD-SMOTE achieves leading or competitive scores.

Ablation studies, provided in Appendix M, indicate that using the different model components, namely, the metric learning loss, importance oversampling, and baseline-classifier filtering, are beneficial for the oversampling task.

## 5 CONCLUDING REMARKS, LIMITATIONS, AND FUTURE WORK

This work explores the potential of using autoencoders in a supervised manner for minority oversampling. We presented ML-AE - an autoencoder trained with a class-reweighted metric learning loss over its latent space. In addition, we proposed a new approach for filtering synthetic samples using a baseline tabular classifier and an adaptation of an importance-oversampling scheme for the AE framework. Combining these elements results in the proposed model named TD-SMOTE.

The proposed oversampling technique was evaluated over 36 datasets, achieving competitive, and in many cases leading, score ranks (AP, F1, and ROC-AUC) for both downstream classifications with Catboost and SVM classifiers.

The model poses two main limitations. The first is the scalability of the importance-oversampling component. The oversampling importance of each minority sample is determined, as presented in Algorithm 3, on the KNN algorithm which is not scalable with growing dataset sizes (Weber et al., 1998; Pestov, 2013). The second model limitation is that the filtering baseline classifier must be able to estimate the probability distribution of a sample to belong to each one of the classes.

Future works may include a simple extension for the multi-class setting, the adoption of more advanced tabular-oriented architectures, such as SAINT (Somepalli et al., 2021) and TabNet (Arik & Pfister, 2019), exploring additional filtering schemes based on pre-trained tabular classifiers and scalable importance estimation algorithms.

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

## A   DEEP SMOTE WITH PERMUTATION LOSS RESULTS

Table 3 demonstrates the performance of deep smote (Dablain et al., 2021) which uses a proposed intra-class permutation loss for improving the quality of the generated synthetic samples. The model was adjusted to tabular data by replacing convolutional layers with fully-connected ones.

Table 3: Classification AP results using CatBoost (Deep SMOTE)

|  | smote | bsmote-1 | bsmote-2 | random oversampling | loss reweight | deep smote |
|---|---|---|---|---|---|---|
| glass-0-1-6_vs_2 | 0.439 | 0.383 | 0.455 | 0.379 | 0.375 | 0.341 |
| glass2 | 0.275 | 0.256 | 0.331 | 0.237 | 0.35 | 0.288 |
| glass4 | 0.7 | 0.466 | 1 | 0.638 | 0.805 | 0.411 |
| page-blocks-1-3_vs_4 | 0.966 | 0.966 | 1 | 1 | 1 | 1 |
| yeast-0-5-6-7-9_vs_4 | 0.737 | 0.682 | 0.707 | 0.781 | 0.673 | 0.445 |
| yeast-1_vs_7 | 0.224 | 0.350 | 0.360 | 0.311 | 0.287 | 0.24 |
| yeast-1-2-8-9_vs_7 | 0.201 | 0.213 | 0.208 | 0.207 | 0.215 | 0.124 |
| yeast-1-4-5-8_vs_7 | 0.371 | 0.453 | 0.300 | 0.281 | 0.273 | 0.127 |
| yeast-2_vs_4 | 0.948 | 0.900 | 0.861 | 0.911 | 0.945 | 0.877 |
| yeast-2_vs_8 | 0.527 | 0.316 | 0.232 | 0.532 | 0.521 | 0.548 |
| yeast4 | 0.385 | 0.324 | 0.264 | 0.431 | 0.393 | 0.347 |
| yeast5 | 0.8 | 0.877 | 0.927 | 0.848 | 0.814 | 0.869 |
| yeast6 | 0.733 | 0.649 | 0.681 | 0.700 | 0.754 | 0.135 |
| AP Mean | 0.562 | 0.526 | 0.563 | 0.558 | 0.570 | **0.442** |

## B    FEATURE PREPROCESSING AND REPRESENTATION

Tabular datasets may include both numerical and categorical features. In addition, they may include missing values. The first step involves imputing missing values. Numerical features are imputed with the mean value of the feature, and categorical features are represented by a 0-vector (instead of a one-hot vector). Continuous and discrete numeric features are scaled using a standard scaling per feature, calculated over the training set. More complex numerical representations such as mode-specific normalization(Xu et al., 2019) demonstrated marginal to no improvement. Categorical features are one-hot encoded as it is a more meaningful representation for representing the different categories (line 3 in Algorithm 1 and line 5 in Algorithm 2). Some tabular classifiers (e.g., Catboost) support categorical features. For this reason, the synthetically generated minority samples should be represented with categorical values similar to the original data. Hence, the fields associated with the one-hot encodings in the synthetic samples are translated back to categories according to the maximal value (line 9 in Algorithm 2).

## C    ML-AE ARCHITECTURE

One architecture was used over all datasets for simplicity and proof of concept. This approach is common in many deep tabular models, such as CTGAN(Xu et al., 2019) and TTGAN(Gradstein et al., 2022). The sole hyperparameter of the architecture is the latent space dimension, denoted by $|z|$. Adjusting the number of layers and widths per dataset (i.e., hyperparameters) may improve the results.

$$
E_\phi(x) = \begin{cases}
h_0 = LReLU_{0.2}\big(FC_{|x|\to 32\cdot|x|}(x)\big) \\
h_1 = LReLU_{0.2}\big(FC_{32\cdot|x|\to 16\cdot|x|}(h_0)\big) \\
z = FC_{16\cdot|x|\to|z|}(h_1)
\end{cases}
$$

$$
D_\psi(z) = \begin{cases}
h_0 = SiLU\big(FC_{|z|\to 8\cdot|x|}(z)\big) \\
h_1 = SiLU\big(FC_{8\cdot|x|\to 16\cdot|x|}(h_0)\big) \\
h_2 = SiLU\big(FC_{16\cdot|x|\to 32\cdot|x|}(h_1)\big) \\
\hat{x}_i = FC_{32\cdot|x|\to 1}(h_2) & i \in N_n \\
\hat{x}_i = SoftMax(FC_{32\cdot|x|\to m_i}(h_2)) & i \in N_c
\end{cases}
$$

Notations:

1. $FC$ - fully connected layer.

2. SiLU(Ramachandran et al., 2017) and LReLU - the non-linear activations.

3. $N_n$ and $N_c$ - the indices of numeric and categorical features, respectively.

4. $m_i$ - the number of categories in the $i$'th categorical feature.

## D    METRIC LEARNING ILLUSTRATIONS

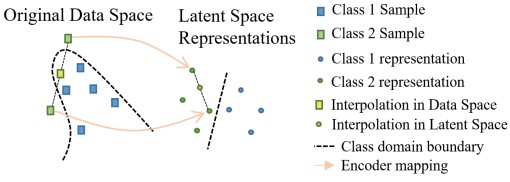

Figure 4: Class separation example - original vs. latent spaces

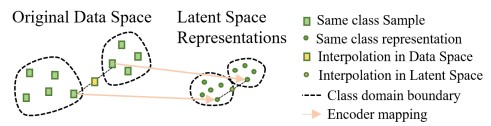

Figure 5: Class density example - original vs. latent spaces

# E GETIMPORTANCE ILLUSTRATION

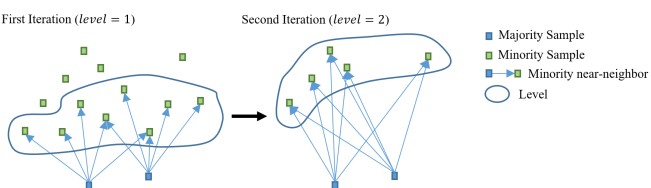

Figure 6: GetImportance (Algorithm 3) illustration - in each iteration, the $k$-NNs minority samples of each majority sample are found (blue arrows). These minority samples are assigned the iteration index, which reflects the proximity level to majority points, and removed from the set of minority samples used in the next iteration. In this manner, each minority sample is assigned a proximity level by which its importance is determined.

# F WEIGHTED SMOTE ALGORITHM

---

**Algorithm 4** WeightedSMOTE

---

**Inputs**: $x^{min}$ - minority samples,
  $weights$ - importance-weights,
  $num\_required$ - number of required samples
**Outputs**: interpolations,
  $base\_indices$ - interpolation bases,
  $neighbors\_indices$ - interpolation neighbors

1: $k \leftarrow 5$                                                                                      ▷ default number of $k$-NNs is 5
2: $k\_neighbors \leftarrow k$ nearest-neighbors of $x^{min}$ from $x^{min}$                ▷ returns $|x^{min}| \times k$ matrix
3: $base\_indices \leftarrow$ choose $num\_required$ samples from $[0, ..., |x^{min}| - 1]$ w.p. $weights$
4: $neighbors\_indices \leftarrow$ uniformly sample $num\_required$ neighbor indices from $[0, ..., k - 1]$
5: $\lambda \leftarrow$ choose $num\_required$ samples from a $Uniform[0, 1]$ probability
6: $x^{base} \leftarrow x^{min}[base\_indices]$
7: $x^{neighbors} \leftarrow k\_neighbors[base\_indices][neighbors\_indices]$
8: $interpolations \leftarrow x^{base} + \lambda(x^{base} - x^{neighbors})$
9: **return** $interpolations, base\_indices, neighbors\_indices$

---

# G FILTERING ALGORITHM

The filtering algorithm is detailed in Algo.5. A visual demonstration of the filtering is provided in Fig.7

---

**Algorithm 5** Filter

**Inputs**: trained baseline classifier- $classifier$, synthetic samples- $x_{syn}$, interpolation factors- $\lambda$, original data- $x$, base samples- $base\_indices$, neighbor samples- $nhbr\_indices$

**Output**: filtered $x_{syn}$

1: $base\_score \leftarrow classifier(x[base\_indices])$
2: $nhbr\_score \leftarrow classifier(x[nhbr\_indices])$
3: $syn\_score \leftarrow classifier(x_{syn})$
4: $thr\_score \leftarrow \frac{(base\_score)^{1-\lambda}(nhbr\_score)^{\lambda}}{margin}$
5: **return** $x_{syn}[syn\_score \geq thr\_score]$

---

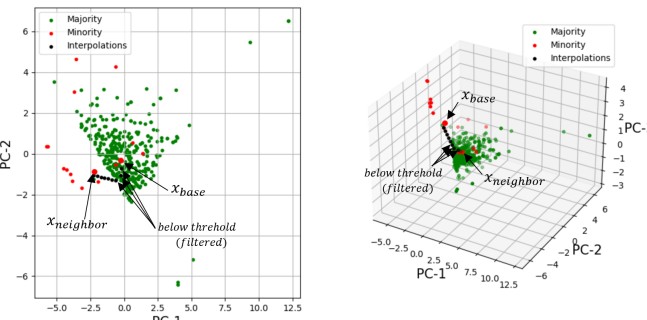

Figure 7: Example - yeast 2-vs-8 dataset - the figures show the 2D & 3D PCAs of the ML-AE latent space. The bold red dots marked by $x_{base}$ and $x_{neighbor}$ are a random pair of minority near-neighbors. Nine evenly-spaced interpolations between the latent samples are marked with black dots. These interpolations are decoded and filtered using a pre-trained Catboost classifier with the proposed threshold. Three decoded interpolations got very low scores for belonging to the minority class ($\sim 0.01$) and are filtered. Not surprisingly, they seem to be positioned close to the majority domain as opposed to the rest of the interpolations.

## H  TOY EXAMPLE

Simple synthetic datasets may exemplify the benefits of TD-SMOTE for minority oversampling. For this purpose, a synthetic 2D class imbalanced dataset was generated. The *circles* dataset, illustrated in Fig.8, is sampled from a multi-circle shaped distribution where each class is associated with distinct radii.

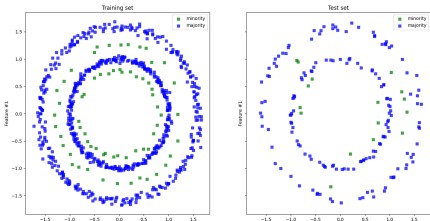

Figure 8: *Circles* dataset - both minority and majority samples are generated with evenly spaced angles and radii sampled from a bi-normal distribution (two circles per class). The minority and majority radii are sampled from - $0.5 \cdot \mathcal{N}(0.8, 0.03) + 0.5 \cdot \mathcal{N}(1.28, 0.03)$ and $0.5 \cdot \mathcal{N}(1, 0.03) + 0.5 \cdot \mathcal{N}(1.6, 0.03)$, respectively. The class imbalance ratio is set to be $0.1$.

The *circle* dataset is oversampled using SMOTE and TD-SMOTE. The oversampled datasets, illustrated in Fig.9, clearly demonstrates how SMOTE generates synthetic samples that may cross into

the majority domain, a phenomenon that appears less frequent in the oversampled with TD-SMOTE dataset.

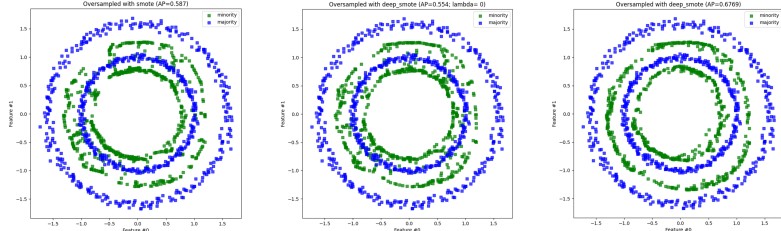

Figure 9: *Circles* oversampled dataset - both SMOTE (left) and TD-SMOTE without metric learning loss ($\lambda_{metric} = 0$, middle) generate samples in the inner circle of the majority distribution. This does not occur in the TD-SMOTE oversampling (right). TD-SMOTE achieves the highest AP in a downstream classification with Catboost compared to other baseline oversampling techniques.

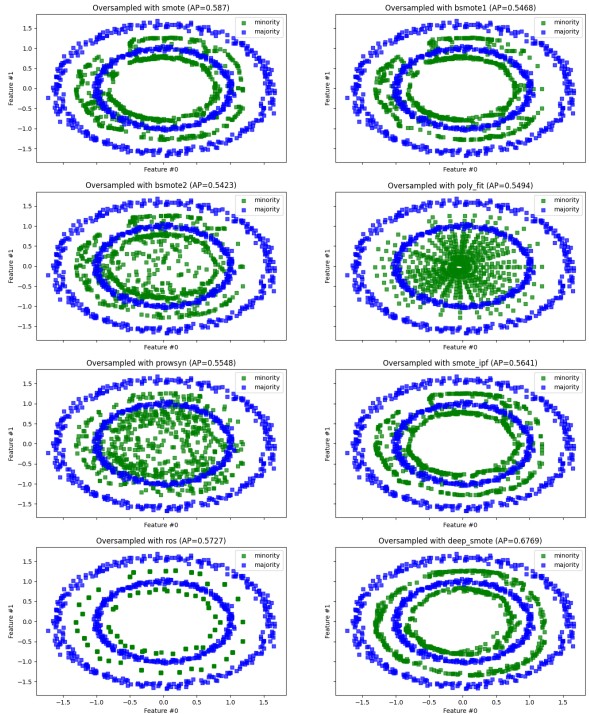

Figure 10: *Circles* Dataset - oversampling with different schemes. TD-SMOTE achieves the highst AP score for downstream classification with Catboost.

## I    EVALUATION

Evaluation of oversampling methods is done according to the quality of a downstream classification task using the oversampled data. The following section elaborates on the different facets of the implemented evaluation process.

### I.1    PERFORMANCE MEASURE

The most widely used empirical measure for classifier training and performance evaluation is accuracy. Accuracy does not distinguish between the number of correct labels from different classes,

which may lead to erroneous conclusions in the case of imbalanced datasets. For example, in extreme class imbalance cases, a classifier that always predicts the majority class will have high accuracy, not reflecting its erroneous predictions over the minority class. In addition to accuracy, there are several other performance metrics, such as precision, recall, and false-positive-rate (FPR), which are defined as:

$$Accuracy := \frac{TP+TN}{P+N}, \quad Precision := \frac{TP}{TP+FP},$$
$$Recall := \frac{TP}{TP+FN}, \quad FPR := \frac{FP}{FP+TN} = \frac{FP}{N}.$$

Where: $T$, $F$, $P$, and $N$ represent True, False, Positive, and Negative, respectively. These measures are informative in class-imbalance cases and are the basis of the actual measures that are used for evaluation (e.g., AP). In the scope of this work, evaluation is done using three measures commonly used for classification tasks over class-imbalanced datasets (More, 2016; Johnson & Khoshgoftaar, 2019):

1. Average Precision (AP) - a numeric approximation of the Precision-Recall plot AUC.

2. F1 Score = $2 \cdot \frac{Precision \cdot Recall}{Precision+Recall}$ with a decision threshold of 0.5.

3. ROC-AUC - a numeric approximation of the Recall-FPR plot AUC. $FPR$ is based, by definition, on the number of negatives ($N$). For imbalanced datasets, $N$ may be substantially larger than False-Positive ($PN$), rendering the changes in $PN$ to have less of an impact over the final ROC-AUC. Nevertheless, this metric will also be presented in the results, as it is commonly used in previous research (More, 2016; Darabi & Elor, 2021).

## I.2 OVERSAMPLING STRATEGY

Oversampling is done to achieve a 1:1 ratio between classes. This is the default oversampling strategy for the library baselines (e.g., SMOTE). Other approaches may include lower oversampling rates combined with undersampling of the majority class. Because the experiments are focused on comparing the oversampling methods and not the sampling strategy, all experiments are conducted under a 1:1 ratio oversampling strategy.

## I.3 DOWN-STREAM CLASSIFIERS

As of this research, the gradient-boosting decision-tree family is the go-to model for tabular classifications (Shwartz-Ziv & Armon, 2021; Kadra et al., 2021a). One of the best classifiers in this family, and overall, is CatBoost(Prokhorenkova et al., 2017; Ibrahim et al., 2020). For this reason, CatBoost was used as the main downstream classifier in the experiments. Its parameter values are specified in appendix-J.2. In the research field of oversampling methods, it is well documented that the performances of oversampling methods are classifier dependent (Bej et al., 2021a; More, 2016). Comparisons are also conducted with an SVM classifier to align with previous research.

## I.4 BASELINES

TD-SMOTE is compared to a set of oversampling techniques and generative models.

1. Generative Models One may hypothesize that using a good tabular generative model may be a straightforward and simple solution for minority oversampling. The generative model is trained to learn the data distribution (either solely of the minority samples or of the entire data) and then used to generate new synthetic minority samples. To check the validity of this hypothesis, a state-of-the-art tabular generative model called CTGAN (Xu et al., 2019) is added as an additional baseline for comparison. In the scope of this work, CTGAN is trained over all the data by representing the class as an additional categorical feature. At inference, the model is conditioned to generate samples from the minority class. It is worth mentioning that the CTGAN paper presents good results when evaluated on class-imbalanced datasets, rendering it a viable option without special adaptations. For further details regarding hyperparameter selection, see - appendix J.1.

2. Datasets with Categories Modern and state-of-the-art tabular classifiers such as CatBoost support categorical features. For this reason, in the scope of this work, oversampling tech-

niques are expected to generate synthetic samples with categorical values when generating a categorical feature. Some oversampling methods do not support categorical features as they can not generate samples with categorical values. For this reason, evaluation over purely numerical datasets and mixed/fully categorical datasets is done with different oversampling methods. Purely-numeric dataset oversampling is evaluated with loss-reweighting, random oversampling, and the common and high-performing SMOTE variations (Section 2), CTGAN (Xu et al., 2019) and the proposed model TD-SMOTE. Datasets with categories are evaluated with loss-reweighting, random oversampling, SMOTE-NC, SMOTE-N, CTGAN, and TD-SMOTE.

## I.5 HYPERPARAMETERS

The choice of hyperparameters can significantly affect the experiment results. This is evident in previous works and our experiments as well (More, 2016; Darabi & Elor, 2021). A persistent underlying question is how much effort was put into optimizing the proposed model compared to the alternative baselines. To alleviate this issue, the number of hyperparameters in the proposed model was intentionally intended to be low. For example, a single encoder-decoder architecture is used for all datasets. Assuming no prior or domain knowledge, the hyperparameters are chosen (solely) according to their 5-fold cross-validation score over the training set. For further details, including hyperparameters and search ranges, see appendix J.1.

## I.6 STATISTICAL SIGNIFICANCE

The Friedman test is performed over all methods to assess the statistical significance, followed by a post hoc analysis based on the Wilcoxon signed rank test (for pairwise comparisons). These tests are a standard metric for comparing classifiers across multiple datasets (Demšar, 2006; Bej et al., 2019; Tarawneh et al., 2020; Kadra et al., 2021b). The Friedman test compares all the methods. The $\mathcal{H}_0$ hypothesis is that all methods are similar, while the $\mathcal{H}_1$ hypothesis is that at least one method is superior to some other method. The $\mathcal{H}_0$ hypothesis of the Wilcoxson signed rank test is that the observations of the two methods are similar, while the $\mathcal{H}_1$ hypothesis is that one is superior to the other.

## I.7 DATASETS

A set of 36 datasets with high class-imbalance ratios are used for evaluation. The dataset size, imbalance ratio, and the number of features are specified in Table 4. The datasets are publicly available under the Keel[4] and Imblearn[5] libraries. In the case of the Webpage and Arrhythmia datasets, the high number of features (300 and 278, respectively) resulted in the SVM classifier training failing; hence, these two datasets are excluded in experiments using SVM.

---

[4] https://sci2s.ugr.es/keel/imbalanced.php?order=ins#sub30
[5] https://imbalanced-learn.org/stable/datasets/index.html

Table 4: Dataset Characteristics

|  | Dataset | #Samples | Imbalance Ratio | #Features (Categorical) |
|---|---|---|---|---|
| Purely Numeric | Glass-0-1-6_vs_2 | 192 | 10.3 | 9 (0) |
|  | Glass2 | 214 | 11.6 | 9 (0) |
|  | Glass4 | 214 | 15.5 | 9 (0) |
|  | Page-Blocks-1-3_vs_4 | 472 | 15.9 | 10 (0) |
|  | Yeast-0-5-6-7-9_vs_4 | 528 | 9.4 | 8 (0) |
|  | Yeast-1_vs_7 | 459 | 14.3 | 7(0) |
|  | Yeast-1-2-8-9_vs_7 | 947 | 30.6 | 8(0) |
|  | Yeast-1-4-5-8_vs_7 | 693 | 22.1 | 8 (0) |
|  | Yeast-2_vs_4 | 514 | 9.1 | 8 (0) |
|  | Yeast-2_vs_8 | 482 | 23.1 | 8 (0) |
|  | Yeast4 | 1484 | 28.1 | 8 (0) |
|  | Yeast5 | 1484 | 32.7 | 8 (0) |
|  | Yeast6 | 1484 | 41.4 | 8 (0) |
|  | Ecoli | 336 | 8.6 | 7(0) |
|  | Letter Image | 20,000 | 26.5 | 16 (0) |
|  | Libras Move | 360 | 14.0 | 90 (0) |
|  | Mammography | 11,183 | 42.2 | 6 (0) |
|  | Ozone Levels | 2536 | 34.0 | 72 (0) |
|  | Pen Digits | 10,992 | 9.5 | 16 (0) |
|  | Satimage | 6,435 | 9.3 | 36 (0) |
|  | Spectrometer | 531 | 10.8 | 93 (0) |
|  | US Crime | 1,994 | 12.3 | 100 (0) |
|  | Webpage | 34,780 | 35.0 | 300 (0) |
|  | Wine Quality | 4,898 | 15.9 | 85 (0) |
|  | Yeast ME2 | 1,484 | 28.1 | 8 (0) |
|  | Yeast ML8 | 2,417 | 12.6 | 103 (0) |
|  | Coil 2000 | 9,822 | 15.9 | 85 (0) |
|  | Oil Spill | 937 | 21.9 | 49 (0) |
|  | Optical Digits | 5,620 | 9.2 | 64 (0) |
|  | Arrhythmia | 452 | 17.1 | 278 (0) |
|  | Car Evaluation | 1728 | 12.0 | 21 (0) |
| With Categories | Abalone | 4177 | 9.7 | 8 (1) |
|  | Abalone19 | 4177 | 129.5 | 8 (1) |
|  | Sick Euthyroid | 3163 | 9.8 | 24 (18) |
|  | Thyroid Sick | 3772 | 15.3 | 28 (21) |
|  | Solar Flare | 1389 | 19.4 | 10 (10) |

## J  HYPERPARAMETER SELECTION APPENDIX

### J.1  BASELINES

Table 5: Hyperparameter selection

| Model | Hyperparameter | Values | Note |
|---|---|---|---|
| TDSMOTE | latent dimension size | 0.75 | ML-AE latent space dimension relative to the number of features in the original data space. |
| | $\lambda_{metric}$ | 1 | The factor of $\mathcal{L}_{metric-learn}$ in $\mathcal{L}_{tot}$. |
| SMOTE  Chawla  et  al. (2002) | n_neighbors | 3, 5, 7 | Number of nearest neighbors to interpolate with. |
| Borderline-SMOTE  Han et al. (2005) | n_neighbors | 3, 5, 7 | Number of nearest neighbors to interpolate with. |
| | m_neighbors | 5, 10 | Number of nearest neighbors according to which to determine type. |
| PolyFit  Gazzah  &  ES-SOUKRI  BEN  AMARA (2008) | - | - | The star topology is chosen as best candidate according to previous results. |
| ProWSyn  Barua  et  al. (2013b) | L | 3, 5, 7 | Number of proximity levels. |
| | n_neighbors | 3, 5, 7 | Number of nearest neighbors according to which to determine type. |
| SMOTE-IPF  Sáez  et  al. (2015) | n_folds | 5, 9 | Number of folds when filtering (see paper). |
| | n_neighbors | 3, 5, 7 | Number of nearest neighbors to interpolate with. |
| SMOTENC Chawla et al. (2002) | n_neighbors | 3, 5, 7 | Number of nearest neighbors to interpolate with. |
| SMOTEN  Chawla  et  al. (2002) | - | - | |
| Xu et al. (2019) CTGAN | embedding_dim | 16, 32, 64 | Embedding dim (normal distribution from which latent is. sampled) |
| | batch_size | 8, 16, 32 | |
| | pac | 8, 16 | See details in PAC-GAN paper Lin et al. (2018) |
| | epochs | 100, 150 | |

### J.2  CATBOOST PARAMETERS

100 iterations (trees), learning rate=0.2, depth of each tree = 6.
Rest are default - loss = NLL.
Auto class reweight is used when class reweighting is checked.

# K ADDITIONAL PERFORMANCE MEASURES APPENDIX

## K.1 F1 RESULTS

Table 6: Classification F1 results for numerical datasets using CatBoost

| | no_os | smote | bsmote1 | bsmote2 | poly_fit | prowsyn | smote_ipf | ros | rw | ctgan | td-smote |
|---|---|---|---|---|---|---|---|---|---|---|---|
| glass-0-1-6_vs_2 | 0.000 | 0.333 | 0.286 | 0.286 | 0.286 | 0.333 | 0.286 | 0.333 | 0.333 | 0.286 | **0.400** |
| glass2 | 0.000 | 0.333 | 0.364 | **0.400** | 0.000 | 0.222 | 0.364 | 0.364 | 0.364 | 0.222 | 0.286 |
| glass4 | 0.667 | 0.667 | 0.667 | 0.667 | 0.667 | 0.667 | 0.667 | 0.667 | **0.750** | 0.500 | 0.667 |
| page-blocks-1-3_vs_4 | 0.800 | 0.833 | **0.909** | **0.909** | 0.833 | 0.833 | **0.909** | 0.833 | **0.909** | 0.889 | 0.800 |
| yeast-0-5-6-7-9_vs_4 | 0.500 | **0.818** | 0.727 | 0.783 | 0.500 | 0.741 | 0.783 | 0.727 | 0.727 | 0.286 | 0.400 |
| yeast-1_vs_7 | 0.250 | 0.286 | **0.400** | 0.250 | 0.000 | 0.222 | 0.267 | 0.308 | 0.267 | 0.000 | 0.250 |
| yeast-1-2-8-9_vs_7 | **0.250** | 0.143 | 0.143 | 0.133 | 0.143 | 0.118 | 0.133 | 0.200 | 0.167 | **0.250** | 0.222 |
| yeast-1-4-5-8_vs_7 | 0.000 | 0.286 | **0.500** | 0.375 | 0.000 | 0.400 | 0.364 | 0.182 | 0.286 | 0.000 | 0.000 |
| yeast-2_vs_4 | 0.857 | 0.846 | 0.720 | 0.769 | 0.833 | 0.846 | 0.846 | 0.857 | 0.833 | **0.957** | 0.880 |
| yeast-2_vs_8 | **0.667** | 0.571 | 0.500 | 0.400 | **0.667** | 0.571 | 0.571 | **0.667** | **0.667** | **0.667** | **0.667** |
| yeast4 | 0.286 | **0.414** | 0.320 | 0.294 | 0.286 | 0.378 | 0.353 | 0.364 | 0.364 | 0.154 | 0.400 |
| yeast5 | 0.824 | 0.842 | **0.889** | 0.643 | **0.889** | 0.800 | 0.842 | 0.842 | 0.750 | 0.842 | **0.889** |
| yeast6 | 0.667 | 0.588 | 0.500 | 0.632 | 0.615 | 0.571 | 0.556 | 0.533 | 0.600 | 0.667 | **0.714** |
| ecoli | **0.588** | 0.417 | 0.417 | 0.370 | 0.471 | 0.414 | 0.435 | 0.444 | 0.462 | 0.471 | 0.471 |
| letter_img | **0.972** | 0.966 | 0.960 | 0.945 | 0.958 | 0.955 | 0.963 | 0.960 | 0.942 | 0.951 | 0.966 |
| libras_move | 0.571 | 0.667 | 0.462 | 0.500 | 0.462 | 0.667 | 0.500 | 0.429 | 0.667 | **0.750** | 0.545 |
| mammography | 0.626 | 0.597 | 0.597 | 0.442 | **0.673** | 0.567 | 0.567 | 0.615 | 0.587 | 0.507 | 0.617 |
| ozone_level | 0.118 | 0.296 | 0.286 | **0.357** | 0.118 | 0.250 | 0.250 | 0.273 | 0.333 | 0.174 | 0.111 |
| pen_digits | **0.998** | **0.998** | 0.984 | 0.979 | 0.991 | 0.993 | 0.995 | **0.998** | 0.995 | 0.988 | 0.995 |
| satimage | 0.669 | 0.641 | 0.635 | 0.642 | 0.688 | 0.642 | **0.691** | 0.636 | 0.605 | 0.609 | 0.658 |
| spectrometer | 0.875 | 0.889 | 0.800 | **0.941** | 0.875 | **0.941** | 0.750 | 0.824 | **0.941** | 0.875 | **0.941** |
| us_crime | 0.200 | 0.393 | 0.339 | 0.345 | 0.279 | **0.400** | 0.364 | 0.392 | 0.310 | 0.158 | **0.400** |
| webpage | 0.661 | 0.543 | 0.583 | 0.635 | 0.658 | 0.655 | 0.552 | 0.409 | 0.400 | 0.657 | **0.672** |
| wine_quality | 0.170 | 0.239 | 0.214 | 0.219 | 0.209 | 0.243 | 0.239 | **0.313** | 0.260 | 0.263 | 0.215 |
| yeast_me2 | 0.533 | 0.540 | 0.438 | 0.381 | 0.533 | 0.450 | 0.500 | 0.471 | 0.516 | **0.625** | 0.588 |
| yeast_ml8 | 0.000 | 0.000 | 0.133 | 0.127 | 0.093 | 0.068 | 0.000 | 0.050 | 0.000 | 0.000 | **0.145** |
| coil_2000 | 0.015 | 0.128 | 0.103 | 0.073 | 0.076 | 0.062 | 0.123 | 0.245 | **0.246** | 0.032 | 0.047 |
| oil | 0.455 | 0.412 | 0.444 | 0.390 | **0.667** | 0.421 | 0.410 | 0.621 | 0.583 | 0.615 | 0.643 |
| optical_digits | 0.967 | 0.977 | 0.977 | 0.973 | 0.959 | 0.977 | 0.977 | 0.968 | **0.986** | 0.967 | 0.982 |
| arrhythmia | **0.909** | **0.909** | **0.909** | 0.769 | 0.800 | **0.909** | 0.833 | **0.909** | 0.833 | **0.909** | **0.909** |
| car_eval_34 | 0.412 | 0.540 | 0.579 | 0.615 | 0.412 | 0.500 | 0.500 | **0.683** | **0.683** | 0.500 | 0.500 |
| #Best | 6 | 4 | 5 | 4 | 4 | 3 | 2 | 5 | 7 | 6 | 9 |
| #Worst | 8 | 1 | 2 | 8 | 5 | 1 | 2 | 1 | 4 | 7 | 3 |

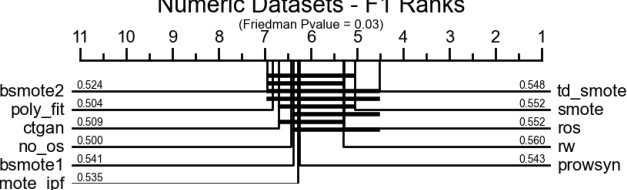

Figure 11: Critical Difference Diagram - Wilcoxson significance analysis over the purely numerical datasets using Catboost. Ranks connected via a bold bar indicate that performances are not significantly different, namely that $p > 0.05$. Average F1 scores are presented above lines.

.

## K.2   ROC-AUC Results

Table 7: Classification ROC results for numerical datasets using CatBoost

| | no_os | smote | bsmote1 | bsmote2 | poly_fit | prowsyn | smote_ipf | ros | rw | ctgan | td-smote |
|---|---|---|---|---|---|---|---|---|---|---|---|
| glass-0-1-6_vs_2 | 0.671 | 0.729 | 0.657 | 0.786 | 0.536 | **0.836** | 0.650 | 0.571 | 0.614 | 0.807 | 0.643 |
| glass2 | 0.718 | 0.660 | 0.795 | 0.744 | 0.750 | 0.737 | 0.692 | 0.654 | **0.827** | 0.795 | 0.699 |
| glass4 | 0.967 | 0.967 | 0.942 | **1.000** | 0.933 | 0.958 | 0.967 | 0.975 | 0.983 | 0.942 | 0.975 |
| page-blocks-1-3_vs_4 | 0.998 | 0.998 | 0.998 | **1.000** | 0.996 | 0.996 | 0.998 | **1.000** | **1.000** | **1.000** | 0.998 |
| yeast-0-5-6-7-9_vs_4 | 0.944 | 0.962 | 0.928 | 0.939 | **0.965** | 0.941 | 0.964 | 0.954 | 0.933 | 0.943 | 0.922 |
| yeast-1_vs_7 | **0.952** | 0.839 | 0.915 | 0.868 | 0.752 | 0.861 | 0.874 | 0.907 | 0.897 | 0.950 | 0.930 |
| yeast-1-2-8-9_vs_7 | 0.611 | 0.568 | 0.653 | 0.634 | 0.593 | 0.601 | 0.595 | 0.600 | 0.629 | 0.707 | **0.744** |
| yeast-1-4-5-8_vs_7 | 0.585 | **0.811** | 0.737 | 0.706 | 0.712 | 0.758 | 0.792 | 0.743 | 0.663 | 0.556 | 0.723 |
| yeast-2_vs_4 | 0.993 | 0.993 | 0.985 | 0.981 | 0.986 | 0.992 | 0.989 | 0.985 | 0.992 | **0.997** | 0.992 |
| yeast-2_vs_8 | 0.618 | 0.680 | 0.642 | 0.613 | 0.664 | 0.653 | 0.583 | 0.691 | 0.581 | **0.745** | 0.723 |
| yeast4 | 0.932 | 0.934 | 0.916 | 0.883 | 0.944 | 0.936 | 0.930 | 0.931 | 0.934 | 0.953 | **0.957** |
| yeast5 | 0.995 | 0.994 | 0.996 | 0.998 | 0.992 | 0.995 | 0.996 | 0.996 | 0.996 | **0.998** | 0.996 |
| yeast6 | 0.970 | 0.960 | 0.980 | 0.979 | 0.982 | 0.974 | 0.954 | 0.974 | 0.986 | **0.988** | 0.971 |
| ecoli | 0.883 | 0.876 | 0.848 | 0.836 | 0.917 | 0.871 | 0.888 | 0.900 | 0.895 | **0.936** | 0.912 |
| letter_img | 0.998 | 0.999 | 0.997 | 0.998 | 0.999 | 0.999 | 0.999 | 1.000 | 1.000 | 0.998 | 0.999 |
| libras_move | **0.970** | 0.773 | 0.887 | 0.675 | 0.839 | 0.851 | 0.815 | 0.660 | 0.812 | 0.943 | 0.731 |
| mammography | 0.943 | 0.916 | 0.944 | 0.937 | 0.951 | 0.930 | 0.936 | 0.917 | 0.916 | **0.953** | 0.937 |
| ozone_level | 0.906 | 0.874 | 0.819 | 0.862 | **0.935** | 0.909 | 0.906 | 0.917 | 0.917 | 0.890 | 0.887 |
| pen_digits | **1.000** | **1.000** | **1.000** | 1.000 | **1.000** | **1.000** | **1.000** | **1.000** | **1.000** | 1.000 | **1.000** |
| satimage | 0.948 | 0.949 | 0.948 | 0.950 | 0.952 | **0.955** | 0.952 | 0.951 | 0.947 | 0.933 | 0.947 |
| spectrometer | 0.999 | 0.997 | 0.997 | **1.000** | 0.999 | **1.000** | 0.992 | 0.998 | 0.998 | 0.999 | **1.000** |
| us_crime | 0.882 | 0.878 | 0.883 | 0.875 | **0.890** | 0.887 | 0.855 | 0.882 | 0.861 | 0.864 | 0.886 |
| webpage | 0.961 | 0.963 | 0.960 | 0.957 | 0.956 | 0.959 | 0.964 | 0.967 | **0.968** | 0.964 | 0.956 |
| wine_quality | 0.804 | 0.806 | 0.816 | 0.770 | 0.761 | 0.795 | 0.820 | 0.815 | **0.828** | 0.805 | 0.819 |
| yeast_me2 | 0.941 | **0.972** | 0.912 | 0.933 | 0.949 | 0.968 | 0.972 | 0.970 | 0.972 | 0.922 | 0.934 |
| yeast_ml8 | 0.606 | 0.584 | 0.558 | **0.612** | 0.592 | 0.581 | 0.595 | 0.596 | 0.511 | 0.555 | 0.541 |
| coil_2000 | 0.766 | 0.753 | 0.745 | 0.738 | 0.766 | **0.767** | 0.748 | 0.736 | 0.758 | 0.754 | 0.755 |
| oil | 0.954 | 0.948 | 0.948 | 0.935 | 0.977 | 0.951 | 0.940 | 0.980 | 0.959 | 0.965 | **0.981** |
| optical_digits | **1.000** | 0.998 | 0.999 | 0.995 | 0.999 | 0.999 | 0.999 | 0.997 | 0.999 | 0.999 | 0.999 |
| arrhythmia | **0.993** | 0.991 | **0.993** | 0.981 | 0.991 | **0.993** | 0.991 | 0.991 | **0.993** | 0.991 | **0.993** |
| car_eval_34 | 0.954 | 0.999 | 0.995 | 0.996 | 0.988 | **1.000** | 1.000 | 0.999 | 0.984 | 0.971 | 0.984 |
| #Best | 5 | 3 | 2 | 4 | 4 | 7 | 1 | 3 | 6 | 7 | 6 |
| #Worst | 1 | 2 | 3 | 7 | 7 | 1 | 3 | 3 | 3 | 3 | 1 |

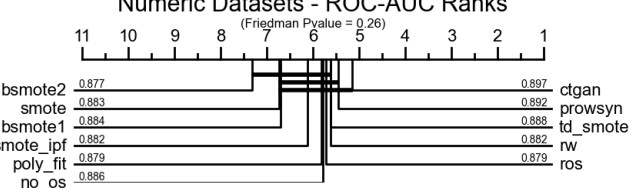

Figure 12: Critical Difference Diagram - Wilcoxson significance analysis over the purely numerical datasets using Catboost. Ranks connected via a bold bar indicate that performances are not significantly different, namely that $p > 0.05$. Average ROC-AUC scores are presented above lines.

## L   SVM Down-Stream Classifier

Table 8: Classification AP results for numerical datasets using SVM

| | no oversample | smote | bsmote1 | bsmote2 | poly_fit | prowsyn | smote_ipf | random oversample | loss reweight | ctgan | td-smote |
|---|---|---|---|---|---|---|---|---|---|---|---|
| glass-0-1-6_vs_2 | 0.164 | 0.410 | 0.408 | **0.654** | 0.472 | 0.530 | 0.402 | 0.431 | 0.486 | 0.172 | 0.428 |
| glass2 | 0.283 | 0.444 | 0.307 | 0.291 | **0.513** | 0.442 | 0.444 | 0.444 | 0.399 | 0.091 | 0.370 |
| glass4 | 0.756 | 0.639 | 0.639 | **1.000** | 0.639 | 0.639 | 0.639 | 0.639 | 0.917 | 0.319 | 0.806 |
| page-blocks-1-3_vs_4 | **1.000** | 0.596 | 0.967 | 0.927 | 0.927 | 0.810 | 0.596 | 0.596 | 0.686 | 0.925 | 0.629 |
| yeast-0-5-6-7-9_vs_4 | **0.650** | 0.599 | 0.585 | 0.488 | 0.575 | 0.616 | 0.609 | 0.575 | 0.575 | 0.414 | 0.647 |
| yeast-1_vs_7 | **0.609** | 0.345 | 0.259 | 0.257 | 0.255 | 0.232 | 0.359 | 0.301 | 0.316 | 0.253 | 0.409 |
| yeast-1-2-8-9_vs_7 | **0.237** | 0.104 | 0.146 | 0.138 | 0.064 | 0.079 | 0.128 | 0.114 | 0.157 | 0.213 | 0.095 |
| yeast-1-4-5-8_vs_7 | 0.239 | 0.307 | 0.363 | **0.430** | 0.388 | 0.392 | 0.298 | 0.299 | 0.280 | 0.097 | 0.313 |
| yeast-2_vs_4 | **0.963** | 0.908 | 0.874 | 0.751 | 0.927 | 0.915 | 0.926 | 0.916 | 0.916 | 0.944 | 0.906 |
| yeast-2_vs_8 | **0.548** | 0.444 | 0.186 | 0.135 | 0.534 | 0.527 | 0.529 | 0.527 | 0.443 | 0.273 | 0.388 |
| yeast4 | 0.442 | 0.354 | 0.323 | 0.247 | 0.316 | 0.324 | 0.352 | 0.267 | 0.284 | 0.238 | **0.451** |
| yeast5 | **0.815** | 0.758 | 0.764 | 0.718 | 0.769 | 0.764 | 0.764 | 0.762 | 0.706 | 0.500 | 0.781 |
| yeast6 | 0.660 | 0.618 | 0.591 | 0.666 | **0.723** | 0.683 | 0.655 | 0.600 | 0.643 | 0.339 | 0.686 |
| ecoli | 0.496 | **0.527** | 0.363 | 0.247 | 0.432 | 0.405 | 0.426 | 0.521 | 0.507 | 0.271 | 0.438 |
| letter_img | 0.990 | 0.996 | 0.995 | 0.861 | **0.997** | 0.992 | 0.996 | 0.995 | 0.993 | 0.919 | 0.996 |
| libras_move | 0.863 | 0.856 | 0.856 | 0.733 | 0.794 | 0.722 | 0.853 | 0.863 | **1.000** | 0.883 | 0.846 |
| mammography | **0.634** | 0.575 | 0.536 | 0.291 | 0.581 | 0.560 | 0.579 | 0.567 | 0.545 | 0.507 | 0.509 |
| ozone_level | 0.218 | 0.268 | 0.240 | 0.130 | 0.256 | 0.242 | 0.216 | 0.279 | **0.291** | 0.180 | 0.257 |
| pen_digits | **1.000** | **1.000** | 1.000 | 0.998 | 1.000 | 1.000 | **1.000** | **1.000** | 1.000 | 0.998 | **1.000** |
| satimage | 0.457 | 0.437 | 0.402 | 0.553 | 0.512 | 0.469 | 0.484 | 0.509 | 0.669 | 0.451 | **0.731** |
| spectrometer | **1.000** | **1.000** | **1.000** | **1.000** | 0.989 | **1.000** | **1.000** | **1.000** | **1.000** | **1.000** | **1.000** |
| us_crime | 0.445 | 0.418 | 0.413 | 0.315 | 0.461 | **0.492** | 0.418 | 0.438 | 0.421 | 0.482 | 0.420 |
| wine_quality | **0.167** | 0.110 | 0.132 | 0.080 | 0.090 | 0.096 | 0.104 | 0.112 | 0.111 | 0.143 | 0.138 |
| yeast_me2 | 0.617 | 0.650 | 0.584 | 0.523 | 0.556 | 0.545 | **0.670** | 0.641 | 0.627 | 0.422 | 0.647 |
| yeast_ml8 | **0.138** | 0.107 | 0.116 | 0.130 | 0.122 | 0.103 | 0.106 | 0.102 | 0.101 | 0.080 | 0.101 |
| coil_2000 | 0.128 | 0.133 | 0.134 | 0.125 | 0.135 | 0.145 | **0.152** | 0.138 | 0.142 | 0.089 | 0.144 |
| oil | 0.520 | 0.774 | 0.788 | 0.835 | 0.723 | 0.768 | 0.763 | 0.821 | **0.887** | 0.813 | 0.762 |
| optical_digits | **1.000** | 1.000 | 0.998 | 0.997 | **1.000** | 1.000 | 1.000 | 0.999 | 0.999 | 0.995 | 0.999 |
| car_eval_34 | **1.000** | **1.000** | **1.000** | **1.000** | **1.000** | **1.000** | **1.000** | **1.000** | **1.000** | 0.345 | **1.000** |
| #Best | 14 | 4 | 2 | 5 | 5 | 3 | 5 | 3 | 5 | 1 | 5 |
| #Worst | 2 | 1 | 1 | 8 | 2 | 2 | 1 | 1 | 0 | 13 | 0 |

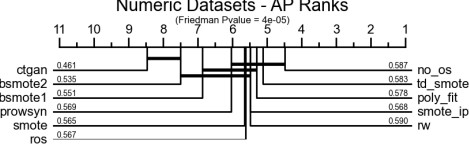

Figure 13: Critical Difference Diagram - Wilcoxson significance analysis over the purely numerical datasets using SVM. Ranks connected via a bold bar indicate that performances are not significantly different, namely that $p > 0.05$. Average AP scores are presented above lines.

.

## M   Ablation Studies

Ablation tests were done for downstream classification with Catboost and evaluation with AP values. The tests were done over all datasets relative to the standard TD-SMOTE training and model (with all its components). The first ablation tests were conducted to examine the ML-AE training methodology and presented in Table 9.

| Training | w/o majority | w/o loss reweight | w/o metric learn loss |
|---|---|---|---|
| Performance | -15.7% | -2.1% | -20.1% |

Table 9: Training Ablation

The tests clearly demonstrate that using the majority samples is beneficial when training the ML-AE. This was demonstrated in previous works, such as TAEI(Darabi & Elor, 2021), and reaffirmed in our tests. In addition, it is evident that both a loss reweighted by class sizes and adding a metric learning loss that uses the labels to shape the latent space are beneficial.

The second set of tests explores the contribution of importance oversampling (i.e., "importance") and classifier filtering (i.e., "filtering"). They are presented in Table 10.

| Oversample | w/o importance | w/o filtering | w/o importance & filtering |
|---|---|---|---|
| Performance | -3.9% | -3.3% | -12.0% |

Table 10: Oversample Ablation

The results demonstrate that the combination of importance-oversampling and classifier filtering is beneficial. Importance weights are determined by proximities to the majority samples. Hence, high-weighted minority samples are near class-domain borders, which inherently increases the hazard of crossing class domains when interpolating. For this reason, it is not surprising that filtering synthetic samples is even more necessary and effective when such importance-oversampling is used.

A central aspect of TD-SMOTE is the hypothesis that oversampling in a learned latent space shaped by a metric-learning loss is more suited for oversampling with SMOTE (Section 3.1.1). To evaluate this hypothesis, a TD-SMOTE version that does not apply importance-oversampling or filtering, which we name Plain-TD-SMOTE, is compared to SMOTE. Recall that TD-SMOTE oversamples in the latent space using SMOTE; hence, the Plain-TD-SMOTE version differs from SMOTE only in the space where the oversampling is done. In such a comparison, the Plain-TD-SMOTE got a better rank, as shown in Fig. 14. The complete set of results is provided in Fig. 11. In this case, improving over SMOTE supports the hypothesis that oversampling in the learned latent space results in better samples for downstream classification.

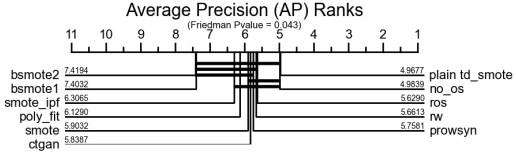

Figure 14: Critical Difference Diagram - Plain-TD-SMOTE
.

Table 11: Classification AP results for numerical datasets using CatBoost - SMOTE vs. Plain-TD-SMOTE

|  | smote | plain-td-smote |
|---|---|---|
| glass-0-1-6_vs_2 | 0.434 | **0.445** |
| glass2 | 0.234 | **0.377** |
| glass4 | **0.809** | 0.589 |
| page-blocks-1-3_vs_4 | **0.967** | **0.967** |
| yeast-0-5-6-7-9_vs_4 | **0.783** | 0.648 |
| yeast-1_vs_7 | **0.586** | 0.407 |
| yeast-1-2-8-9_vs_7 | 0.209 | **0.253** |
| yeast-1-4-5-8_vs_7 | 0.085 | **0.141** |
| yeast-2_vs_4 | 0.925 | **0.927** |
| yeast-2_vs_8 | 0.524 | **0.533** |
| yeast4 | 0.417 | **0.498** |
| yeast5 | 0.905 | **0.932** |
| yeast6 | 0.722 | **0.785** |
| ecoli | **0.564** | 0.483 |
| letter_img | 0.981 | **0.986** |
| libras_move | **0.798** | 0.642 |
| mammography | **0.699** | 0.674 |
| ozone_level | **0.351** | 0.260 |
| pen_digits | **1.000** | **1.000** |
| satimage | 0.755 | **0.777** |
| spectrometer | **0.989** | **0.989** |
| us_crime | 0.388 | **0.432** |
| webpage | **0.698** | 0.685 |
| wine_quality | **0.323** | 0.174 |
| yeast_me2 | 0.601 | **0.641** |
| yeast_ml8 | **0.138** | 0.091 |
| coil_2000 | 0.169 | **0.198** |
| oil | 0.347 | **0.812** |
| optical_digits | **0.996** | 0.995 |
| arrhythmia | **0.877** | 0.810 |
| car_eval_34 | 0.746 | **0.894** |

