# OpenReview forum: "Tabular Deep-SMOTE: A supervised autoencoder-based minority-oversampling technique for class-imbalanced tabular classification"
_ICLR.cc/2024/Conference — ICLR 2024 Conference Withdrawn Submission_

### Official Review · Reviewer_KYqz · 2023-10-31

**Soundness:** 2 fair
**Presentation:** 2 fair
**Contribution:** 2 fair
**Rating:** 3
**Confidence:** 3

**Summary:**

The paper proposes a new method for minority oversampling with autoencoders to tackle the class imbalance of tabular data. The proposed method trains an autoencoder with class-reweighted metric learning loss and filters the synthetic samples afterward. They conducted extensive experiments across 36 datasets and showed that the proposed method achieves competitive performance.

**Strengths:**

- The paper is well-written and easy to follow.
- The proposed method is simple and achieves competitive results.

**Weaknesses:**

My major concern is that there are not enough ablation studies nor analysis of the proposed method:
- Can authors provide some insights or ablation on why the Softmax loss is not directly applied to the original latent space but rather a linear-mapped space? What if using a non-linear mapping? How would the performance change if using larger or smaller weights for this loss?
- The authors claim that Normalized Softmax loss performs the best. It would be more convincing and insightful if comparisons with other losses were provided.
- Similarly, it would be great to also see the ablation of the class reweighting factors in Section 3.1.1, and the importance of the proposed oversampling in Section 3.2.1
- What alternative thresholding strategy could be used in Section 3.2.2? Some comparison is needed to demonstrate the effectiveness of the proposed thresholding IMHO. Besides, it would be interesting to compare this with an oracle experiment, where the minority classifier is trained on a large amount of balanced data.

Other concerns:
- In my understanding, the proposed method can be used in broader scenarios. Why is the tabular data the scope of the paper? Is there any part of the method that is specifically designed for tabular data, or explores the structure of the data?
- Is there a better way to rank the methods compared in Table 1? Imagine that we have methods A and B and two datasets,  A outperforms B with a large margin on one dataset but is outperformed by B on another dataset with a small margin, it will count as 1 best for each method, though in this case, A would be the preferred one.

**Questions:**

See weaknesses

---

### Official Review · Reviewer_Uo1S · 2023-10-31

**Soundness:** 2 fair
**Presentation:** 1 poor
**Contribution:** 1 poor
**Rating:** 3
**Confidence:** 5

**Summary:**

The authors proposed Tabular Deep-SMOTE for class-imbalance mitigation in tabular datasets. The oversampling takes place in the latent space of an autoencoder trained in a supervised manner using metric-learning loss. Further, the authors introduce an importance-oversampling scheme that prioritizes oversampling near class domain boundaries. Lastly, to guarantee the quality of synthetic samples, they propose a synthetic sample filtering scheme based on the decision boundary of a pre-trained classifier.

**Strengths:**

Considers an important problem in the community.

**Weaknesses:**

While the authors have reviewed a fair amount of literature, a few significant works that are highly related are missing. For example, the methodology in [1] is very similar in spirit to what has been proposed.

Further, I have failed to understand why the authors limited the experiments only to tabular datasets. Is there something specific to the method that prevents the method from being applied to an image dataset? If not, then I would highly recommend and appreciate benchmarking using standard image datasets.

Moreover, I suggest the usage of evaluation metrics best suited for imbalance learning as proposed in [2].

The authors do not report standard deviations of the performance evaluation. Therefore, it is not clear whether the performance boost is significant. For example, in thyroid_sick dataset the difference in performance w.r.t the runner-up method is 0.002.

A severe limitation of the proposed method is that the datasets used for experimentation are a binary classification problem. So how will it perform in multi-class settings?

Based on the discussion above, my initial opinion is borderline reject.

1.	Mondal, Arnab Kumar, et al. "Minority Oversampling for Imbalanced Data via Class-Preserving Regularized Auto-Encoders." International Conference on Artificial Intelligence and Statistics. PMLR, 2023.
2.	Mullick, Sankha Subhra, et al. "Appropriateness of performance indices for imbalanced data classification: An analysis." Pattern Recognition 102 (2020): 107197.

**Questions:**

1. The paper describes the method as an algorithm without any justification on why should their design choices matter. One can think of establishing generalization bounds, as in [1].

1a. For instance, why should the imposition of a metric loss on the latent space lead to better oversampling (mathematically).

1b. SMOTE and its variants are known to perform extremely badly in high-dimensions (due to their dependence on distance metrics). Therefore why should WeightedSMOTE be any good?

2. The datasets used are very toy-like and not suited for practical scenarios.

3. As described earlier, I don't understand why should the method be restricted to tabular data.

4. The baselines used are obsolete.

5. The writing is very cumbersome (not meticulous).

6. Figure 3 is unreadable and frivolous in my opinion.

In summary,  this paper lacks novelty, has weak experiments, and has bad writing. It needs a lot of improvement before it can be considered in atop venue.

---

### Official Review · Reviewer_RWeJ · 2023-11-01

**Soundness:** 2 fair
**Presentation:** 2 fair
**Contribution:** 1 poor
**Rating:** 3
**Confidence:** 2

**Summary:**

- This paper introduces a new oversampling method “Tabular Deep-SMOTE” which    is applying autoencoder to SMOTE.
- This paper shows experimentally superior AP and AUC score than other oversampling method by adopting three new methods : metric-learning loss, importance sampling in SMOTE , and filtering scheme.

**Strengths:**

Strengths:

- Improved the performance by introducing new oversampling method.

**Weaknesses:**

Weaknesses:

1. metric-learning loss : The paper uses a linear weight for each label, but it doesn't specify whether this weight is a learnable parameter or if it's defined differently. It seems to be a learnable parameter, but if that's the case, I'm not sure why this weight is helpful for distinguish the two classes.

2. cosine similarity : In the latent space, the metric-learning loss is used to ensure good class separation. However, I wonder how well the cosine similarity works. It would be helpful if additional experimental results related to this are provided. Moreover, if the separation is done well by the cosie similarity, there might be no need to use SMOTE.

3. Experiment : The author reviewed the methods of applying the autoencoder to SMOTE. It would be great if numerical results of these methods are listed in the experiments. Also in addition to the number of times the best performance was achieved, it would be helpful if the authors also provide the average ranking.

4. Theoretical result : There is no theoretical justification of the proposed method.

<Minor error>

1. In Algorithm 1, it seems to be indexing the feature "". It would be beneficial to define it before using it.

2. In Algorithm3, the weight seems smaller the closer it is to the major class. It may need correction. Also, adding k_neighbors or to the index doesn't look good. Although it might complicate things, it would be better to use clear definitions.

**Questions:**

NA

---

### Official Review · Reviewer_iTrN · 2023-11-02

**Soundness:** 3 good
**Presentation:** 4 excellent
**Contribution:** 3 good
**Rating:** 3
**Confidence:** 3

**Summary:**

The paper Tabular Deep-SMOTE proposes a new minority oversampling scheme for dealing with imbalanced data, consisting of three components: an auto-encoder trained including class-labels, a novel importance weighting scheme, and filtering synthetic samples using a baseline classifier.
The paper shows that TD-Smote improves over baseline approaches in aggregate using average ranks over datasets.

**Strengths:**

The paper does a very careful analysis of the problem and runs extensive benchmarks with a variety of metrics and learners.
The paper is well-written and the discussion of the relevant literature is comprehensive. Including a GAN-based approach in addition to the traditional approaches is also laudable.

**Weaknesses:**

The very careful analysis presented in the paper seems to confirm my initial intuition, which is that oversampling does not help. Figure 3, Numeric Datasets shows that no oversampling is nearly identical (and most approaches are statistically indistinguishable). Figure 12 and 13 seem to reinforce this reading of the results.
Figure 11 shows improved results in ranks (though all methods are statistically identical), though I would argue that the F1 score is not an appropriate score in this setting, and AUC or AP should be used.
For categorical datasets, there seems to be an advantage to oversampling, in particular TD-Smote (though again, statistically indistinguishable from no oversampling). It would be interesting to investigate why here oversampling seems to be much more helpful than in the numeric case. Unfortunately I did not find the SVM results or the AUC results for categorical and mixed datasets.

Another weakness is that there is no direct comparison against DEAGO and TAEI, which TD-SMOTE is claiming to improve over. While it's obviously not feasible to compare to all relevant methods, these two seem very close, and in particular the paper claims that adding the label to the autoencoder is useful, which is only shown in Table 9 - and which has a somewhat surprising -20% influence. This seems a very big difference given how close the methods are in general, and I'd love to understand the meaning of that number better.

Minor issues:
The citation for ctgan should list NeurIPS as venue, I think.

**Questions:**

Can you explain the numbers in appendix M? Are these average relative changes? The influences seem quite big, given how close all the methods are and how close they often are to "no oversampling".

Was there a particular reason not to include DAEGO and TAEI in the comparison?